# PRAME Is an Effective Tool for the Diagnosis of Nevus-Associated Cutaneous Melanoma

**DOI:** 10.3390/cancers16020278

**Published:** 2024-01-09

**Authors:** Andrea Ronchi, Gerardo Cazzato, Giuseppe Ingravallo, Giuseppe D’Abbronzo, Giuseppe Argenziano, Elvira Moscarella, Gabriella Brancaccio, Renato Franco

**Affiliations:** 1Pathology Unit, Department of Mental and Physical Health and Preventive Medicine, University of Campania “Luigi Vanvitelli”, 80138 Naples, Italy; andrea.ronchi@unicampania.it (A.R.); dabbronzogiuseppe@gmail.com (G.D.); 2Section of Pathology, Department of Precision and Regenerative Medicine and Ionian Area (DiMePRe-J), University of Bari “Aldo Moro”, 70125 Bari, Italy; gerycazzato@hotmail.it (G.C.); giuseppe.ingravallo@uniba.it (G.I.); 3Dermatology Unit, Department of Mental and Physical Health and Preventive Medicine, University of Campania “Luigi Vanvitelli”, 80138 Naples, Italy; giuseppe.argenziano@unicampania.it (G.A.); elvira.moscarella@unicampania.it (E.M.); gabriella.brancaccio@unicampania.it (G.B.)

**Keywords:** melanoma, nevus-associated melanoma, PRAME, immunohistochemistry, dermatopathology

## Abstract

**Simple Summary:**

Nevus-associated cutaneous melanoma (CM) is a malignant melanocytic proliferation occurring in the context of a pre-existing benign melanocytic nevus. This type of melanoma is not infrequent, accounting for up to 35% of all superficial spreading CMs. Discriminating between malignant and benign cellular populations may be challenging, but it is mandatory for diagnostic and staging purposes and is consequently critical for accurate patient management. The aim of this study was to assess the diagnostic performance of PRAME immunohistochemistry in this clinical setting. In this study, PRAME immunohistochemistry showed a perfect specificity and a good sensitivity, showing negative values for all nevus components and positive values for 59.4% of the malignant cellular components. The diagnostic agreement between morphology and PRAME immunohistochemistry was perfect for the diagnosis of nevus components and fair for the diagnosis of malignant components. PRAME was significantly more expressed in thick invasive CMs than in thin cases (*p* = 0.02). Our results support the use of PRAME immunohistochemistry as a useful tool in the diagnosis of nevus-associated CM.

**Abstract:**

(1) Background: Nevus-associated cutaneous melanoma (CM) is relatively common in the clinical practice of dermatopathologists. The correct diagnosis and staging of nevus-associated cutaneous melanoma (CM) mainly relies on the correct discrimination between benign and malignant cells. Recently, PRAME has emerged as a promising immunohistochemical marker of malignant melanocytes. (2) Methods: PRAME immunohistochemistry (IHC) was performed in 69 cases of nevus-associated CMs. Its expression was evaluated using a score ranging from 0 to 4+ based on the percentage of melanocytic cells with a nuclear expression. PRAME IHC sensitivity, specificity, positive predictive values, and negative predictive values were assessed. Furthermore, the agreement between morphological data and PRAME expression was evaluated for the diagnosis of melanoma components and nevus components. (3) Results: PRAME IHC showed a sensitivity of 59%, a specificity of 100%, a positive predictive value of 100%, and a negative predictive value of 71%. The diagnostic agreement between morphology and PRAME IHC was fair (Cohen’s Kappa: 0.3); the diagnostic agreement regarding the benign nevus components associated with CM was perfect (Cohen’s Kappa: 1.0). PRAME was significantly more expressed in thick invasive CMs than in thin cases (*p* = 0.02). (4) Conclusions: PRAME IHC should be considered for the diagnostic evaluation of nevus-associated CM and is most useful in cases of thick melanomas. Pathologists should carefully consider that a PRAME-positive cellular population within the context of a nevus could indicate a CM associated with the nevus. A negative result does not rule out this possibility.

## 1. Introduction

Diagnosing melanocytic neoplasms remains an intricate challenge within the realm of diagnostic pathology. Even with the growing understanding of the molecular features of melanocytic tumors, morphological findings remain the primary means of diagnosis, with molecular biology and immunohistochemistry playing supporting roles [1,2]. Consequently, the pathologist’s expertise plays an important role in both assessing cutaneous melanoma (CM) and identifying essential prognostic factors, with frequent critical results reported in the literture due to the lack of a robust consensus among operators [3,4]. The incidence of CMs arising from a common nevus (nevus-associated CMs) is not infrequent in clinical practice, with up to 35% of superficial spreading CMs being associated with common nevi in some studies [5,6]. Nevus-associated CMs are frequently situated on the trunk and extremities, with superficial spreading melanomas (SSMs) emerging as the predominant histological subtype [7]. The associated nevus is mainly constituted by an intradermal component, although junctional or compound remnants should be plausible [7]. Notably, nevus-associated CM is characterized as a low cumulative sun damage (CSD) CM, with SSM being the most prevalent histological type, particularly on the trunk in patients with high counts of total nevi. In contrast to de novo CM, nevus-associated CM tends to exhibit reduced thickness and a higher frequency of regression [8,9,10].

The diagnosis of CM and the identification of its histological prognostic markers are extremely difficult tasks in cases of nevus-associated CM. Indeed, the presence of a benign cellular population, potentially constituting the majority of the lesion, could obscure the malignant component, leading to potential misinterpretation. Conversely, precise differentiation between malignant and benign cellular populations is imperative for accurate Breslow’s thickness assessments and subsequent effective clinical management. In recent years, PRAME (PReferentially Expressed Antigen of MEelanoma) has emerged as a novel immunohistochemical marker that is useful for discriminating CM from benign nevi [11]. PRAME, a component of the ubiquitin–protein ligase complex, participates in several cellular processes, such as apoptosis, differentiation, growth regulation, transcription, and ubiquitination [12,13]. Its expression spans multiple epithelial and non-epithelial neoplasms, encompassing CM, endometrial carcinomas, ovarian carcinomas, adenoid cystic carcinomas, seminomas, thymic carcinomas, synovial sarcomas, myxoid liposarcomas, and neuroblastomas [14]. Furthermore, PRAME has garnered interest as a potential target for immunotherapy, with ongoing clinical trials exploring its application in cancer treatment [15,16,17]. 

In several studies, PRAME has been used as a malignancy biomarker in melanocytic proliferation, being critical for challenging diagnoses of CM [18,19,20]. While PRAME has been swiftly incorporated into immunohistochemical panels worldwide, comprehensive data regarding its sensitivity and specificity for diagnosing CM remain incomplete. Notably, only one study has previously assessed the efficacy of PRAME in identifying malignant components in nevus-associated CMs, reporting a sensitivity of 67% and a specificity of 100% [21]. In this study, which includes a wider sample of nevus-associated CMs, we assessed a specific range of PRAME immunohistochemical positivity to ascertain sensitivity, specificity, positive predictive values (PPVs), negative predictive values (NPVs), and the diagnostic concordance between IHC and morphological evaluations within this clinic-pathological context.

Thus, the primary objective of this study was to comprehensively evaluate the diagnostic performance of PRAME immunohistochemical positivity in the context of nevus-associated CM, shedding light on its sensitivity and specificity in this particular clinical setting.

## 2. Materials and Methods

### 2.1. Cases Selection

All cases diagnosed as nevus-associated cutaneous melanoma (CM) between 1 January 2021 and 30 June 2023 were collected from the archives of the Pathology Units at “Luigi Vanvitelli” University Hospital in Naples, Italy and “Aldo Moro” University Hospital in Bari, Italy. Notably, PRAME immunohistochemistry was introduced as a routine marker for CM diagnosis in these institutions beginning 1 January 2023. During this period, a total of 315 CM cases were diagnosed, with 75 of them identified as nevus-associated CMs. At the time of the skin biopsies, written consent, including permission to utilize diagnostic data for scientific purposes, was obtained from each patient. Two seasoned dermatopathologists meticulously reviewed all cases, considering both hematoxylin and eosin-stained slides and immunohistochemical slides (HMB45, p16, Ki67) when applicable. The diagnoses for all cases were realized on the basis of a complete agreement between the experienced dermatopathologists. Eventual diagnostic discrepancies were resolved in a joint analysis. The inclusion criteria comprised the following: (1) confirmation of nevus-associated CM diagnosis by both dermatopathologists; (2) availability of formalin-fixed and paraffin-embedded (FFPE) tissue blocks; (3) accessibility to clinical information; (4) possession of informed consent. Sixty-nine cases met all inclusion criteria and were consequently included in the study.

### 2.2. PRAME Immunohistochemistry

Immunohistochemical staining was performed on 5-micron-thick sections obtained from formalin-fixed and paraffin-embedded (FFPE) tissue blocks using the Ventana platform (Ventana BenchMark ULTRA system) following the manufacturer’s instructions. The primary antibody used was anti-PRAME (rabbit monoclonal antibody, Ventana–Roche, clone EPR20330), a prediluted and ready-to-use antibody with a specific concentration of 11.7 μg/mL. Sebaceous glands served as an internal positive control, while non-melanocytic and non-sebaceous cells on each histological slide functioned as internal negative controls. Consistent with previous studies, PRAME expression was assessed using a scoring system ranging from 0 to 4+ [21,22]. The scores were based on the percentage of tumor cells exhibiting immunohistochemical (IHC) nuclear staining. Thus, nuclear staining was recorded as follows: 0 for no nuclear staining, 1+ for the staining of 1–25% of cells, 2+ for the staining of 26–50% of cells, 3+ for the staining of 51–75% of cells, and 4+ when more than 75% of the cells were positive. Scores of 0 and 1+ were considered negative, 2+ was deemed intermediate, and scores of 3+ and 4+ were considered positive (see Table 1). The 3+ score was selected as the optimal threshold to define PRAME positivity due to the fact that it demonstrated higher sensitivity compared to the 4+ score while still maintaining satisfactory specificity in line with recommendations from recent data [23]. Intermediate cases were treated as negative cases for the statistical evaluations. 

### 2.3. Statistical Analysis

Pearson’s *χ*^2^ test was used to establish whether there were any relationships between the frequencies of PRAME positivity and the CM stages in the sample included in this study. Particularly, the relationship was calculated considering differences in PRAME expression between the CM groups. Differences were considered to be significant for values of *p* < 0.05. Our statistical analysis was performed using IBM SPSS Statistic software (Version 29.0.1.0 (171)).

### 2.4. Test Accuracy

Sensitivity, specificity, positive predictive values, and negative predictive values for PRAME IHC were computed. The diagnostic accordance for CM diagnosis between PRAME immunohistochemistry and morphological evaluation (considered the gold standard) was assessed using Cohen’s kappa coefficient.

## 3. Results

### 3.1. Clinical and Pathological Findings

The sample included 69 cases of nevus-associated cutaneous melanoma diagnosed between 1 January 2023 and 30 June 2023. Patients’ ages at the time of diagnosis ranged from 27 to 88 years, with a mean age of 53 years and a median age of 51 years. The cohort included 37 men, resulting in a male-to-female ratio of 1.16:1. Lesions were mainly located on the trunk in 41 out of 69 cases (59.4%), on the inferior limbs in 12 cases (17.4%), on the superior limbs in 9 cases (13.0%), and on the head and neck in the remaining 7 cases (10.2%). The sample consisted of 47 invasive cutaneous melanomas (68.1%) and 22 in situ melanomas (31.9%), all associated with a nevus component. Within the invasive cases, 93.6% were identified as superficial spreading melanoma (SSM). The remaining 6.4% included two melanomas with spitzoid features and one nodular melanoma. A comprehensive summary of the clinical and pathological features of the sample is presented in Table 2.

### 3.2. PRAME Immunohistochemistry and Diagnostic Performance

PRAME immunohistochemical positivity (IHC) yielded reliable results in all cases. For CMs, a score of 0 was assigned to 11 out of 69 cases (15.9%), a score of 1+ to 8 out of 69 cases (11.6%), a score of 2+ to 9 out of 69 cases (13.0%), a score of 3+ to 9 out of 69 cases (13.0%), and a score of 4+ to 32 out of 69 cases (46.4%) (refer to Table 2). Consequently, 19 out of 69 cases (27.6%) were classified as negative, 9 out of 69 cases (13.0%) were classified as intermediate, and 41 out of 69 cases (59.4%) were classified as positive. The negative cases included 11 Superficial Spreading Melanomas (SSMs) and 8 in situ CMs, while the intermediate cases comprised 6 in situ CMs and 3 SSMs. The positive cases consisted of 30 SSMs, 8 in situ CMs, 2 CMs with spitzoid features, and 1 nodular CM (Figure 1). When PRAME findings were analyzed on the basis of Breslow’s thickness to analyze CMs, different results were found in the different stages. Indeed, among the in situ melanomas, totaling 22 cases, 8 cases (36.36%) were positive; in the T1a stage, including 17 cases, 11 (64.7%) were positive; 1 out of 4 T1b cases (25%) was positive; 13 out of 16 stage 2 cases, 6 out of 8 stage 3 cases, and all (2 out of 2) stage 4 cases were PRAME-positive in the neoplastic component. The total amounts of indetermined cases were 1 out of 16 (6.25%) stage 2 cases and 2 out of 8 (25%) stage 3 cases. The remaining cases were negative (Figure 2). 

Notably, the associated melanocytic nevus consistently tested negative in all cases (100%). Representative cases are illustrated in Figure 3, Figure 4 and Figure 5.

### 3.3. Statistical Results

Our statistical analysis showed that thick invasive melanomas (more than T1) are significantly more positive than thin invasive melanomas (T1) (*p* = 0.02). The difference in PRAME expression was not statistically significant when comparing all CM stages (Tis vs. T1 vs. T2 vs. T3 vs. T4) (*p* = 0.38), Tis plus thin CMs vs. thick CMs (*p* = 0.338), and thin non ulcerated CM (T1a) vs. T1b, T2, T3, and T4 CMs (*p* = 0.113).

### 3.4. Test Accuracy

PRAME immunohistochemistry (IHC) for diagnosing cutaneous melanoma (CM) exhibited a sensitivity of 59%, a specificity of 100%, a positive predictive value of 100%, and a negative predictive value of 71%.

The accordance between the morphology and PRAME IHC was deemed fair, with a Cohen’s Kappa coefficient of 0.3. Conversely, the diagnosis of the benign nevus component associated with CM demonstrated perfect agreement, with a Cohen’s Kappa coefficient of 1.0.

## 4. Discussion

Nevus-associated CM is frequently encountered in clinical practice. While most CMs are considered de novo neoplasms, a notable proportion arises from pre-existing nevi [24]. In a recent meta-analysis including 20,126 melanomas from 38 studies, 29.1% were classified as nevus-associated CMs [7]. Within our study sample, nevus-associated CMs accounted for 75 out of 315 cases, representing 23.8% of all cases. Acquired nevi, mainly intradermal nevi (54%), are most commonly found in nevus-associated CM as opposed to junctional (21.7%) or compound (15.4%) remnants [7]. Some evidence indicates that nevus-associated CMs exhibit clinical and histological distinctions compared to de novo CMs. Clinically, the former type of melanoma is more frequently observed in younger age groups and tends to manifest on the trunk and extremities. Histologically, it is characterized by either superficial spreading melanoma (SSM) or nodular melanoma (NM) and is correlated with an early clinical stage and thinner Breslow thickness [6,25,26,27,28,29,30,31,32,33,34,35,36,37,38,39,40,41]. Despite nevus-associated CM being linked to a thinner Breslow thickness, multivariate analyses have shown no significant differences in disease-free survival or overall survival compared to de novo CM [25,26,27].

Dermatologists as well as pathologists experience challenges when managing CMs associated with nevus. When performing routine visits, dermatologists ought to recognize the importance of examining older nevi in order to detect early signs of malignant evolution [7]. Conversely, pathologists face the task of accurately distinguishing between benign and malignant cellular populations within the lesion, providing an accurate diagnosis, and assessing prognostic parameters and lesion stage. Differentiating between these two populations may be challenging, particularly among lesions that have significant regression or are partially hidden by an extensive inflammatory infiltrate. While immunohistochemistry can be helpful, it currently cannot definitively distinguish between the two cell populations. HMB45 immunohistochemistry, often inconclusive, is informative only when the nevus component is deep, and it may yield positive results in junctional and superficial nevi [42]. The potential of p16 immunohistochemistry has been explored for distinguishing between nevus and CM, demonstrating a sensitivity and specificity of 98.0% and 28.6%, respectively [11]. Only one previous study has evaluated the role of PRAME for the evaluation of nevus-associated CM [21]. Lohman et al. examined the outcomes of PRAME immunohistochemistry in a cohort of 36 cases involving nevus-associated CMs. Their findings unveiled a positive PRAME expression in 67% of the evaluated CMs, contrasting starkly with a complete absence of PRAME positivity in the associated nevi [21]. In our study, encompassing a larger set of 69 nevus-associated CMs, we sought to define the diagnostic performance of PRAME immunohistochemistry in differentiating benign and malignant cellular populations. In our study sample, nevus-associated CMs most frequently occurred in adulthood (mean age 53 years) and without significant sex predilection (M:F = 1.16:1), with the most common locations being the trunk (59.4%) and inferior limbs (17.4%). Invasive CMs were prevalent than in situ CMs (68.1% and 31.9%, respectively), with SSM being the most common invasive subtype. Overall, PRAME showed commendable diagnostic performance in distinguishing between benign and malignant cellular populations within the context of nevus-associated CMs. Specifically, PRAME yielded positive results in 59.4% of CM cases and negative results in 27.6% of cases, with a sensitivity of 59% and a negative predictive value (NPV) of 71%. The remaining 13.0% of cases yielded intermediate results (score 2+), necessitating further studies to establish their clinical significance.

In situ melanomas displayed the poorest diagnostic performance, testing positive in 36.3% of cases, negative in 36.3% of cases, and intermediate in 27.4% of cases. These findings differ from those observed by Lohman et al., who reported PRAME positivity in 76% of in situ CM cases [21]. However, the study populations of the existing studies are relatively small, and more data are required to determine the true sensitivity of PRAME immunohistochemistry in this context. It is also plausible that the heterogeneity of results may be partially attributed to the diagnostic challenges in histologically defining in situ CMs. Indeed, the differential diagnosis between in situ CMs and pre-malignant melanocytic proliferations (e.g., severely dysplastic nevus, atypical melanocytic hyperplasia, etc.) is actually burdened by poor reproducibility [43,44,45,46]. The diagnostic agreement among pathologists for in situ CM was found to be 60% in a study conducted by Elmore et al. involving 1187 pathologists from 10 US states [47]. The low positivity rates of in situ CMs may also explain the relatively low Cohen’s Kappa coefficient of 0.3 in our study. PRAME was consistently negative in the nevus populations in all 69 cases, resulting in a specificity of 100% and a positive predictive value (PPV) of 100%. Our statistical analysis revealed a fair agreement between PRAME-based diagnosis and morphology-based diagnosis for CM (Cohen’s Kappa: 0.3) and a perfect agreement for the nevus component diagnosis (Cohen’s Kappa: 1.0). There was no significant statistical correlation between PRAME positivity and Breslow’s thickness. However, our results showed that PRAME is significantly more effective in identifying thick CMs than thin CMs in cases involving a pre-existing nevus (*p* = 0.02). Indeed, PRAME showed positive results in 57.1% of thin (T1 stage) invasive CMs and in 80.8% of thick (t2 plus T3 plus T4) CMs. In this study, PRAME showed positive results in 81.25% of T2 cases, 75% of T3 cases, and 100% of T4 cases, of which there was only two. These results corroborate what was previously reported by Gassenmaier et al., who reported PRAME positivity in 58.6% of thin CMs [48]. We can therefore state that PRAME sensitivity in invasive CMs is greater in thick cases compared to thin ones. This observation is significant because nevus-associated CMs are often thin; therefore, pathologists should keep in mind the different sensitivities of this marker according to the depth of the neoplasm.

This study included two CMs with spitzoid features, both of which tested positive for PRAME IHC. However, PRAME IHC demonstrated poor diagnostic performance when applied to melanocytic neoplasms with spitzoid histology [49,50,51,52]. Nevertheless, the lack of studies on this topic, along with the low number of patients, makes it challenging to draw definitive conclusions. Some studies have included Spitz naevi in various groups, but comparisons are hindered by the incorporation of only single cases of spitzoid melanomas. Gerami et al. reported PRAME positivity in 33% of spitzoid melanomas and in one (2.6%) benign spitzoid tumor [50]. It is crucial to consider that distinguishing between malignant and benign lesions based solely on histology is particularly challenging in the context of spitzoid lesions. A spitzoid morphology does not necessarily imply that the lesion is a Spitz neoplasm [53]. In our study, the two spitzoid CMs did not show tyrosine kinase translocations and were consequently not diagnosed as Spitz melanomas but rather as CMs with spitzoid morphological features. Interestingly, PRAME tested positive in two out of three (66%) malignant cases when considering cases with nodal metastases, as reported by Gerami et al. [50].

## 5. Conclusions

PRAME immunohistochemistry is a useful tool in the diagnostic evaluation of nevus-associated CMs, and the key advantages are its specificity and PPV. Consequently, pathologists should carefully consider that a PRAME-positive cellular population within the context of a nevus could indicate a CM associated with the nevus. Nevertheless, a negative result does not rule out this possibility. In summary, nevus-associated CMs represent a complex issue for both clinicians and pathologists. The integration of PRAME immunohistochemistry shows promise in differentiating between benign and malignant cellular populations, particularly in the context of nevus-associated CMs. However, further studies are warranted to validate these findings and refine the application of PRAME in the diagnosis of melanomas, especially those with spitzoid features.

## Figures and Tables

**Figure 1 cancers-16-00278-f001:**
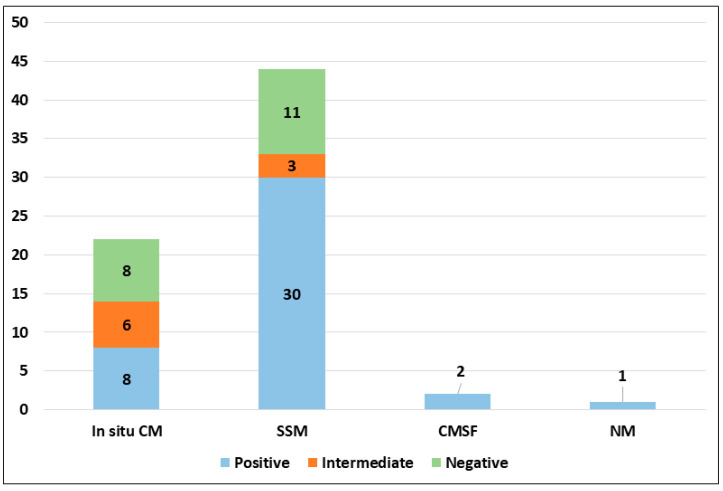
PRAME immunohistochemical expression in malignant population according to histological subtype (abbreviations: CM, cutaneous melanoma; SSM, superficial spreading melanoma; CMSF, cutaneous melanoma with spitzoid features; NM, nodular melanoma).

**Figure 2 cancers-16-00278-f002:**
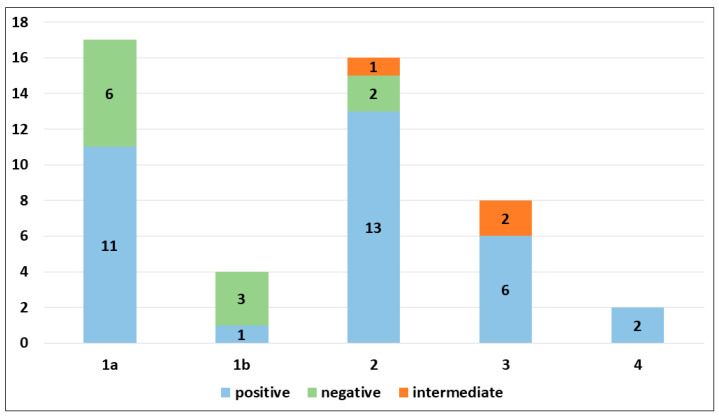
PRAME immunohistochemical expression in invasive cutaneous melanomas according to pathological stage.

**Figure 3 cancers-16-00278-f003:**
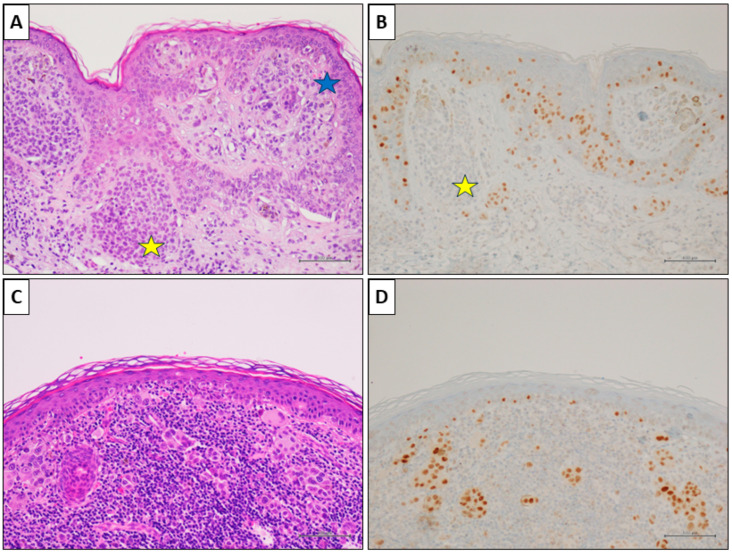
A nevus-associated superficial spreading melanoma from the right arm of a 42-year-old female participant. (**A**) Histological examination showed a in situ and superficial melanoma (blue star); a nevus component is present on the left (yellow star) (H&E, original magnification 100×). (**B**) PRAME IHC marked the melanoma cells (>75%), while it was negative in nevus cells (yellow star) (PRAME immunostain, original magnification 100×). (**C**) In an other field of the same lesion, an invasive epithelioid melanoma with associated intense lymphoid infiltate (H&E, original magnification 100×). (**D**) PRAME IHC marked >75% of the neoplastic cells (PRAME immunostain, original magnification 100×) (scale bar: 100 µm).

**Figure 4 cancers-16-00278-f004:**
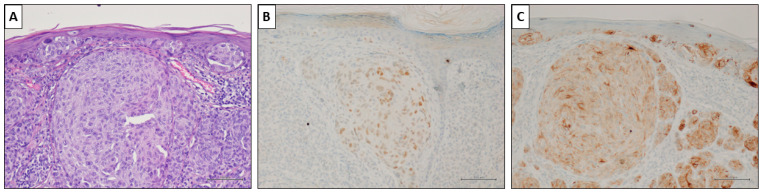
A nevus-associated superficial spreading melanoma from the left arm of a 37-year-old female participant. (**A**) Intra-epithelial and invasive melanoma with a prominent nest in the superficial dermis (H&E, original magnification 200×). (**B**) PRAME marked <50% of malignant cells (PRAME immunostain, original magnification 200×). (**C**) HMB45 showed slightly but diffusely positive values. (HMB45 immunostain, original magnification 200×) (scale bar: 100 µm).

**Figure 5 cancers-16-00278-f005:**
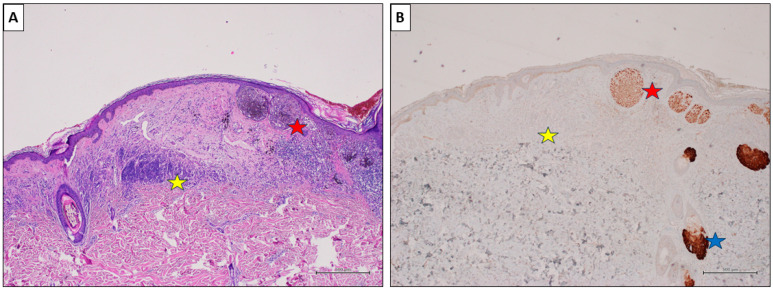
A nevus-associated superficial spreading melanoma from the back of a 54-year-old male participant. (**A**) The melanoma cells are arranged in large nests at the dermo-epidermal junction (red star). On the left, an intradermal nevus component is evident (yellow star) (H&E, original magnification 40×). (**B**) PRAME IHC marked >75% of the malignant cells (red star), while showing negative values in the nevus (yellow star). Sebaceous glands are intensely positive and used as positive controls (blue star) (PRAME immunostain, original magnification 40×) (scale bar: 500 µm).

**Table 1 cancers-16-00278-t001:** PRAME immunohistochemistry scoring.

Score	Positive Cells (%)	Result
0	<1	Negative
1+	1–25	Negative
2+	26–50	Intermediate
3+	51–75	Positive
4+	>75	Positive

**Table 2 cancers-16-00278-t002:** Clinical and pathological findings and immunohistochemical results.

**Age**	
Mean age	53 years
Median age	51 years
Range	27–88 years
**Sex**	
Males	37 (53.6%)
Females	32 (46.4%)
M:F	1.16:1
**Topography**	
Trunk	41 (59.4%)
Inferior limbs	12 (17.4%)
Superior limbs	9 (13.0%)
Head and neck	7 (10.2%)
**Histological Diagnosis**	
SSM	44 (63.8%)
In situ CM	22 (31.9%)
Spitzoid CM	2 (2.9%)
Nodular CM	1 (1.4%)
**PRAME IHC Score in CM**	
Score 0	11 (15.9%)
Score 1+	8 (11.6%)
Score 2+	9 (13.05%)
Score 3+	9 (13.05%)
Score 4+	32 (46.4%)

Abbreviations: SSM, superficial spreading invasive melanoma; CM, cutaneous melanoma; IHC, immunohistochemistry.

## Data Availability

Data are available upon reasonable request. All data needed to evaluate the conclusions of this paper are present in the paper.

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
