# Peer review of "PRAME Is an Effective Tool for the Diagnosis of Nevus-Associated Cutaneous Melanoma"

_cancers, 2024, doi:10.3390/cancers16020278_

Round 1

Reviewer 1 Report

Comments and Suggestions for Authors

Dear Authors,

I've read your paper with great interest and I found the results extremeely important when dealing with nevus associated melanoma.

Comments on the Quality of English Language

english is ok, minor editing is needed

Author Response

We thank the Reviewers for their precious work. We modified the manuscript based on the observations, as following:

Reviewer #1:

I've read your paper with great interest and I found the results extremeely important when dealing with nevus associated melanoma.

Comments on the Quality of English Language: english is ok, minor editing is needed

AA: thank you for your interest in our work. We performed a thorough linguistic revision according to the observation of Reviewer #3.

Reviewer 2 Report

Comments and Suggestions for Authors

The authors use PRAME immunohistochemistry (IHC) prospectively in 69 cases of cutaneous melanoma associated with nevus. They demonstrate that PRAME immunohistochemistry is a useful tool in the diagnosis of BC associated with nevi. The article is well written and the methods are appropriate. However, the following comments should be considered from my point of view to improve the manuscript:

·       The sum of the percentages is not 100 in one of the sections of Table 2 : PRAME IHC score in CM. Score 0 11 (15.9%), Score 1+ 8 (11.6%), Score 2+ 9 (13.0%), Score 3+ 9 (13.0%), Score 4+ 32 (46.4%); The authors should check the rounding

·       The authors should provide an explanation as to why a relatively low Cohen's Kappa of 0.3 has been given for the agreement between PRAME-based diagnosis and morphology-based diagnosis in the diagnosis of CM.

·       Considering that the specificity for the diagnosis shown by the authors is 100% and that however, PRAME positivity has been described in some Spitz nevus (Lezcano et al), and that the authors say that they have 2 cases of melanoma spitzoid, I think this topic should be discussed.

Author Response

We thank the Reviewer for the precious work. we modified the manuscript based on the observations, as following:

Reviewer #2:

The authors use PRAME immunohistochemistry (IHC) prospectively in 69 cases of cutaneous melanoma associated with nevus. They demonstrate that PRAME immunohistochemistry is a useful tool in the diagnosis of BC associated with nevi. The article is well written and the methods are appropriate. However, the following comments should be considered from my point of view to improve the manuscript:

The sum of the percentages is not 100 in one of the sections of Table 2 : PRAME IHC score in CM. Score 0 11 (15.9%), Score 1+ 8 (11.6%), Score 2+ 9 (13.0%), Score 3+ 9 (13.0%), Score 4+ 32 (46.4%); The authors should check the rounding

AA: We corrected the rounding in the table.

The authors should provide an explanation as to why a relatively low Cohen's Kappa of 0.3 has been given for the agreement between PRAME-based diagnosis and morphology-based diagnosis in the diagnosis of CM.

AA: the relatively low K coefficient could be explained by the low PRAME sensitivity for the diagnosis of in situ melanoma in this study. We have expanded the discussion to include comments about this low agreement.

Considering that the specificity for the diagnosis shown by the authors is 100% and that however, PRAME positivity has been described in some Spitz nevus (Lezcano et al), and that the authors say that they have 2 cases of melanoma spitzoid, I think this topic should be discussed.

AA: thank you for your suggestion. We commented this topic in the Discussion.

Reviewer 3 Report

Comments and Suggestions for Authors

In the manuscript entitled “PRAME is an effective tool for the diagnosis of nevus-associated cutaneous melanoma“ the authors aim to verify the diagnostic value of PRAME in the context of nevus-associated cutaneous melanoma. The study is somewhat interesting and deals with an important issue in dermatopathology (i.e. recognition of a CM arising in a nevus), although a similar paper (cited by the authors) was published >2 y ago. The authors should investigate some aspects of their study in more detail to increase novelty of the results. In this reviewer’s opinion a major revision is necessary.

 Please find some more specific comments below:

-        The current study is similar to the one published by Lohman et al. (ref. 17). The authors are encouraged to mention that study in the introduction and highlight why the results provided in that study are in their opinion insufficient.

-        Positivity rate in nevus-associated CM in the study of Lohman et al. was higher, even when only 4+ cases were considered „positive”. How would the authors address this discrepancy?

-        The authors should provide a rationale for considering 3+ and 4+ IHC scores as positive, 2+ as intermediate and 1+ (and 0) as negative.

-        Please add the concentration of the primary antibody.

-        The current study showed that diagnostic performance of PRAME IHC is worse for in situ melanomas than for invasive melanomas. This is contrary to what was observed by Lohman et al. The authors should address this discrepancy in the discussion.

-        Diagnostic accuracy of PRAME IHC would be especially useful in cases where melanoma component (in situ or invasive) is small. Conversely, PRAME IHC would be usually superfluous in those nevus-associated melanomas in which the malignant component is large and evident morphologically. Therefore, it would be of interest to investigate the diagnostic performance of PRAME IHC in relation to the size (area/depth) of malignant component. Is the sensitivity equal (better?) in early, small foci of nevus-associated melanoma or in more advanced tumors? The authors are encouraged to explore this angle in more detail thus increasing novelty of their work.

-      Discussion should be reorganized. Currently, the authors hardly compare their results with previous studies and the last 1/3 of the discussion is essentially repetition of the results.

-        Figure 1 is unnecessary – it takes a lot of space but conveys little information. All data included in Figure 1 are also presented in Figure 2 and in the text.

-        Table 3 partially repeats data that are presented in Figure 1, Figure 2 and in the manuscript text.

-        The authors should mention the use of standard diagnostic IHC stains in Material and Methods section. Otherwise, inclusion of HMB45 immunostain in Figure 4 seems unexpected and out of context, and may be unclear for non-dermatopathologists.

-        The manuscript needs a thorough linguistic revision.

Comments on the Quality of English Language

 The manuscript needs a thorough linguistic revision. Improper wording is used multiple times throughout the text.

Author Response

We thank the Reviewer for the precious work. we modified the manuscript based on the observations, as following:

Reviewer #3:

In the manuscript entitled “PRAME is an effective tool for the diagnosis of nevus-associated cutaneous melanoma“ the authors aim to verify the diagnostic value of PRAME in the context of nevus-associated cutaneous melanoma. The study is somewhat interesting and deals with an important issue in dermatopathology (i.e. recognition of a CM arising in a nevus), although a similar paper (cited by the authors) was published >2 y ago. The authors should investigate some aspects of their study in more detail to increase novelty of the results. In this reviewer’s opinion a major revision is necessary.

 Please find some more specific comments below:

-        The current study is similar to the one published by Lohman et al. (ref. 17). The authors are encouraged to mention that study in the introduction and highlight why the results provided in that study are in their opinion insufficient.

AA: Lohman et al evaluated the daignostic usefeulnes of PRAME IHC in a series of 36 nevus-associated melanomas. However, more data are needed to establish the diagnostic value of PRAME in the different clinical-pathological settings. In our study, we analysed PRAME in a larger series of nevus-associated melanomas, evauating positive predictive value and negative predictive value and diagnostic agreement between PRAME IHC and morphological evaluation. Moreover, in this study we used a different positivity score (3+ rather than 4+), which is more useful in the diagnostic practice, as stated also by a recent metanalysis (Kunc M et al. Diagnostic test accuracy meta-analysis of PRAME in distinguishing primary cutaneous melanomas from benign melanocytic lesions. Histopathology. 2023;83(1):3-14.). So we think that this study may contribute to expand the knoledge about the diagnostic value of PRAME for the diagnosis of nevus-associated melanoma.

Positivity rate in nevus-associated CM in the study of Lohman et al. was higher, even when only 4+ cases were considered „positive”. How would the authors address this discrepancy?

AA: we commented this topic in the Discussion.

The authors should provide a rationale for considering 3+ and 4+ IHC scores as positive, 2+ as intermediate and 1+ (and 0) as negative.

AA: we provided the rationale in Material and Methods section.

Please add the concentration of the primary antibody.

AA: we added the concentration in Material and Methods section.

The current study showed that diagnostic performance of PRAME IHC is worse for in situ melanomas than for invasive melanomas. This is contrary to what was observed by Lohman et al. The authors should address this discrepancy in the discussion.

AA: we stated this discrepancy in the discussion, and have commented these results.

Diagnostic accuracy of PRAME IHC would be especially useful in cases where melanoma component (in situ or invasive) is small. Conversely, PRAME IHC would be usually superfluous in those nevus-associated melanomas in which the malignant component is large and evident morphologically. Therefore, it would be of interest to investigate the diagnostic performance of PRAME IHC in relation to the size (area/depth) of malignant component. Is the sensitivity equal (better?) in early, small foci of nevus-associated melanoma or in more advanced tumors? The authors are encouraged to explore this angle in more detail thus increasing novelty of their work.

AA: we investigate the positivity rate of PRAME according to depth of the melanoma. No significant statistical correlation resulted between depth of melanoma and PRAME positivity. We added a Figure (figure 2) in the Results.

Discussion should be reorganized. Currently, the authors hardly compare their results with previous studies and the last 1/3 of the discussion is essentially repetition of the results.

AA: we expanded and reorganized the Discussion, comparing our results with previous studies and added some topics in the Discussion.

Figure 1 is unnecessary – it takes a lot of space but conveys little information. All data included in Figure 1 are also presented in Figure 2 and in the text.

AA: We deleted the figure.

Table 3 partially repeats data that are presented in Figure 1, Figure 2 and in the manuscript text.

AA: We deleted the table.

The authors should mention the use of standard diagnostic IHC stains in Material and Methods section. Otherwise, inclusion of HMB45 immunostain in Figure 4 seems unexpected and out of context, and may be unclear for non-dermatopathologists.

AA: We mentioned the IHC stains in Material and Methods section.

The manuscript needs a thorough linguistic revision.

AA: we performed a thorough linguistic revision.

Round 2

Reviewer 3 Report

Comments and Suggestions for Authors

Dear Authors,

Thank you for uploading the revised version of your manuscript. In this reviewer’s opinion, the scientific quality of the paper has been significantly improved. However, some flaws still need to be addressed. Please see the remarks below:

-        Thank you for performing additional analyses related to pT stages of studied CMs. However, statistical methods for analysis of PRAME/pT stages are not given. In L174 the Authors state that there was „no significant correlation” – was this truly a correlation analysis? Probably not.

-        Irrespective of statistical significance and tests that were used, PRAME expression does seem to differ between early vs advanced melanomas. In thin (pT1) melanomas PRAME was expressed in 57% of cases, whereas pT2, pT3 and pT4 tumors have 81%, 75% and 100% of positive cases, respectively (or 81% in combined pT1+pT2+pT3). Even if no statistical difference was observed between the groups (if so, likely impacted by low numbers of cases), 57% vs 81% is a notable disparity. It corroborates previous results (e.g. 10.3390/cancers13153864) that show lower sensitivity of PRAME IHC for thin melanomas. Because thin melanomas are highly represented among nevus-associated CMs, this is a caveat for the use of PRAME in this context.

-        ‘CMSF’ is used in Figure 1 (only there) without explanation. Please explain abbreviations or use ‘spitzoid CM’ for consistency.

-        Conclusions at the end of the manuscript „pathologist should carefully consider ....a nevus-associated CM” are more accurate than firm statement in the abstract „PRAME-positive population in the context of a nevus represents a CM”

-        Despite noticeable linguistic corrections, the manuscript still contains some erroneous/awkward-sounding phrases (e.g. L18 „This is a not infrequent occurrence”, L35 „percentage of melanocytic cells nuclear expression”, L129 „due to its demonstrated higher sensitivity”). Moreover, it could be much more consise. Some ideas within the introduction are repetitive and there is some overlap between introduction and discussion. Description of results is too lengthy and replicates what is shown in figures/tables. Some results are unnecessarily presented in the discussion (e.g. LL 252-261).

In this reviewer’s opinion the manuscript would greatly benefit from a more in-depth linguistic/editorial revision.

-        Please make sure that final quality of microphotographs is good. PDF version available for revision contains figures with very poor quality.

Comments on the Quality of English Language

In this reviewer’s opinion the manuscript would greatly benefit from a more in-depth linguistic/editorial revision.

Author Response

Dear Editor and Reviewers,

Thank you for giving me the opportunity to re-submit a revised draft of my manuscript titled PRAME is an effective tool for the diagnosis of nevus-associated cutaneous melanoma to Cancers. We appreciate the time and effort that you and the reviewer have dedicated to providing your valuable feedback on my manuscript. We have been able to incorporate changes to reflect most of the suggestions provided by the reviewers. We have highlighted the changes within the manuscript.

Here is a point-by-point response to the reviewers’ comments and concerns.

1) Thank you for performing additional analyses related to pT stages of studied CMs. However, statistical methods for analysis of PRAME/pT stages are not given. In L174 the Authors state that there was „no significant correlation” – was this truly a correlation analysis? Probably not.

AA: The Pearson's χ2 test was used to establish whether there were any relationships between the frequencies of PRAME positivity and the CM stage groups. We stated it in the Material and methods section. Moreover, we expanded the statistical analysis, analyzing the expression of PRAME on the basis of the different pathological stages. Thanks to your suggestion, we performed a more extensive statistical analysis, showing that PRAME was significantly more expressed in thick than in thin CMs (P=0.02).

2) Irrespective of statistical significance and tests that were used, PRAME expression does seem to differ between early vs advanced melanomas. In thin (pT1) melanomas PRAME was expressed in 57% of cases, whereas pT2, pT3 and pT4 tumors have 81%, 75% and 100% of positive cases, respectively (or 81% in combined pT1+pT2+pT3). Even if no statistical difference was observed between the groups (if so, likely impacted by low numbers of cases), 57% vs 81% is a notable disparity. It corroborates previous results (e.g. 10.3390/cancers13153864) that show lower sensitivity of PRAME IHC for thin melanomas. Because thin melanomas are highly represented among nevus-associated CMs, this is a caveat for the use of PRAME in this context.

AA: thank you for your suggestion. We agree that there is a notably different positivity rate between thin and thick melanomas. So, we performed a more extensive statistical analysis showing that PRAME was significantly more expressed in thick than in thin CMs (P=0.02). We highlighted these findings in the discussion, also commenting about the results of Gassenmaier et al (10.3390/cancers13153864). We also highlited that nevus-assocated melanoma is often thin, and consequently pathologist should keep in mind the different sensitivity of this marker according to the depth of the neoplasm.

3) ‘CMSF’ is used in Figure 1 (only there) without explanation. Please explain abbreviations or use ‘spitzoid CM’ for consistency.

AA: We explained abbreviations in the figure’s legend.

4) Conclusions at the end of the manuscript „pathologist should carefully consider ....a nevus-associated CM” are more accurate than firm statement in the abstract „PRAME-positive population in the context of a nevus represents a CM”.

AA: We corrected accordingly the sentence in both main text and abstract.

5) Despite noticeable linguistic corrections, the manuscript still contains some erroneous/awkward-sounding phrases (e.g. L18 „This is a not infrequent occurrence”, L35 „percentage of melanocytic cells nuclear expression”, L129 „due to its demonstrated higher sensitivity”). Moreover, it could be much more consise. Some ideas within the introduction are repetitive and there is some overlap between introduction and discussion. Description of results is too lengthy and replicates what is shown in figures/tables. Some results are unnecessarily presented in the discussion (e.g. LL 252-261).

AA: We modified the text to reduce repetitions through the manuscript. On the other hand, in order to obtain the required adequate length of the text, we have to preserve the full description of the results, according to  the editorial requirements and as emphasized by the editor in both rounds of revision recommendations.

In this reviewer’s opinion the manuscript would greatly benefit from a more in-depth linguistic/editorial revision.

AA: We performed a linguistic revision thanks to the help of a native speaker.

6) Please make sure that final quality of microphotographs is good. PDF version available for revision contains figures with very poor quality.

AA: We followed the editorial instructions of the journal. Thus, we control that the resolution of the graphics corresponds 1200 dpi and the resolution of the histological images corresponds to 300 dpi, as journal requires to the authors